# Scholars180: An effective oral presentation assessment for optometry students

**Khyber Alam[1], Xinyi Zheng[1] *, Kate Vance[1], Kenneth Lee[2]**

**1** Discipline of Optometry, School of Allied Health, The University of Western Australia, Perth, Western Australia, Australia, **2** Discipline of Pharmacy, School of Allied Health, The University of Western Australia, Perth, Western Australia, Australia

\* xinyizheng14@gmail.com

## Abstract

Oral presentation assessments are multifunctional tools that can potentially test all six cognitive domains of Bloom's taxonomy. Yet, they are not used as frequently as other forms of assessment in curriculums due to time limitations. Hence, designing effective oral presentation assessments that can overcome this is required. The purpose of this study was to investigate whether Scholars180, an oral presentation assessment developed for optometry students, would effectively help students improve their knowledge of and confidence in the identification and management of ocular diseases. This study utilized a non-randomized pre-questionnaire and post-questionnaire design where the participants (n = 31) were asked to assess their knowledge of ocular diseases before and after the oral presentation. The questionnaire was developed according to the unit outcomes. The responses to each of the 12 Likert-type scale questions on the post-questionnaire with the respective responses on the pre-questionnaire were compared. Students (n = 31) experienced improvements in their knowledge of eye diseases and even more so in their confidence and application of their knowledge. This was indicated by the statistically significant increases in median scores and low interquartile ranges (IQR) of ≤1.0. The peer evaluation also illustrated that students felt that the assessment contributed positively to their learning experience. Teachers require a variety of assessment methods to accurately test the student's authentic depth of knowledge and achievement of learning outcomes. Scholars180 is an effective assessment that follows constructive alignment and overcomes time limitations, providing teachers an assessment to consider implementing in the future.

## Introduction

Traditional assessments involve written and oral assessments. Written assessments can be advantageous as they are a familiar tool to test the student's comprehension of content and it is more objective in comparison to oral assessments, which increases their reliability [1]. However, oral assessments should be used to supplement written ones as some learning outcomes are infeasible to be tested in written form [2]. Unlike the static responses given in written assessments, the oral format allows the assessor not only to test verbal communication but to also probe questions to gain a more well-rounded understanding of the student's knowledge

**Data Availability Statement:** All relevant data are within the manuscript and its Supporting information files.

**Funding:** The authors received no specific funding for this work.

**Competing interests:** The authors have declared that no competing interests exist.

and way of thinking [2]. The oral assessment format allows the potential evaluation of all six cognitive domains of Bloom's taxonomy, which are knowledge, comprehension, application, analysis, evaluation, and creation [2]. This makes it a valuable multifunctional assessment type. Additionally, assessments can also be classified into summative and formative assessments [3]. Examples of summative assessments are final exams, standardized tests, and final reports, whilst formative assessments could include homework assignments, weekly quizzes, and in-class discussions. For effective evaluation, formative assessments should be designed to help strengthen summative assessments [3].

The university has the responsibility to ensure graduate optometrists have been equipped with relevant skills before entering the workforce. Optometric competencies include efficiently performing all optometric procedures, recognizing the importance of effective communication, demonstrating the ability to assess and interpret information, contributing to the creation of new knowledge through research, and being a reflective practitioner to ensure growth and development [4]. As accrediting bodies, such as the Optometry Council of Australia and New Zealand (OCANZ), become more strict about accrediting standards, universities are required to demonstrate a clear and methodical approach to curriculum development, learning outcomes, and relevant assessments [5]. Thus, the multifunctional use of oral presentation assessments should make it an attractive assessment type in addition to other forms of assessments to demonstrate the achievement of learning objectives, but this is not always the case. Despite the need for oral presentation assessments, they are not as commonly used as written assessments. One of the major factors that prevent teachers from utilizing it is the long assessment time that is associated with conducting oral presentations [6]. Yet, a challenge that arises when time is reduced is the simultaneous maintenance of the effectiveness of the assessment, which is sufficient time to achieve pre-defined learning outcomes and a showcase of knowledge.

Although there is a range of existing oral assessments available, there were no specific oral presentation assessments that aligned with the purpose of testing the optometry students on their knowledge and management of ocular diseases. To address this issue, The University of Western Australia (UWA) designed Scholars180, which is an oral presentation assessment for first-year optometry students. UWA's optometry course is based on the competency standards for optometrists in Australia and New Zealand. The competency standards include communication, patient examination, diagnosis and management, and health information management. Constructive alignment was used when designing Scholars180 as it was based on the learning outcomes of a clinical unit and the objectives of the learning events in the semester. Thus, Scholars180 can help display student achievement in these competencies.

For Scholars180, students received 12 weeks to prepare a short 3-minute individual oral presentation about an ocular disease to their peers and teacher. This presentation style is based on The University of Queensland's (UQ) Three Minute Thesis (3MT) format [7] with the addition of a Q&A section. The 3MT format not only challenges students to condense a large amount of information into 180 seconds alongside a single static slide, but it also needs to be done in a manner that is easily comprehensible to an audience who are not experts in the field [7]. Through this assessment, students experienced active and peer learning. Therefore, it was hypothesized that students will experience an improvement in their knowledge of ocular diseases and confidence in the identification and management of ocular diseases.

## Methodology

The Scholars180 assessment is an oral presentation that encompasses 4 components:

a. Baseline assessment of knowledge

b. Oral presentation

c. Q&A session

d. Post-presentation 'experiences' questionnaire

The participating students in this study were first-year optometry students and thus recruitment for their participation in the study was conducted at the beginning of the semester. The study was approved by The University of Western Australia (approval no. 2021/ET000652). Written and informed consent were obtained from all participants prior to enrolment in the study. Informed consent was required as participants were asked to conduct tests and the researchers used the data collected. All participants were informed of possible publishing and written consent was obtained. All the first-year optometry students were given the opportunity to participate in this study unless they did not consent to participate. The authors declare they have no conflicts of interest, and this research received no specific grant from any funding agency in the public, commercial, or not-for-profit sectors.

In this study, the student's baseline assessment of knowledge of all the eye conditions listed in Table 1 was assessed prior to the assessment. After 12 weeks, a post-presentation 'experiences' questionnaire was used to compare the changes that students experienced after undergoing the assessment. Additionally, the questionnaires developed were based on the unit's learning outcomes to see if they had been achieved. For the oral presentation component, the students were first allocated an eye disease as per Table 1 in week 1 of the semester as part of their introductory lecture. The diseases or topics listed in Table 1 were based on some of the most prevalent eye conditions around the world. In addition, these topics were linked to specific cases which were scheduled to be taught in the second and third years of the Optometry course. The students then had 12 weeks to develop an oral presentation (only one static slide allowed) to encompass the background knowledge related to the eye disease, pathophysiology, signs and symptoms, prevalence, and treatment options. The problem-based learning (PBL) cases throughout the semester were designed in such a way it fed the students sufficient information they could use to structure their presentations. For the presentation itself, the students had 180 seconds to individually present the eye disease to their peers. The assessment was graded using the rubric found in S1 Appendix and the presentation was worth 7% of the total mark for the unit.

For their presentations, the student was required to follow the 3MT presentation format. The 3MT format restricts students to a 180-second presentation, where they can only use one static slide. The lack of time in the 3MT format forces students to remove jargon to concisely explain the disease. Like how a physician would require in-depth clinical knowledge to identify key clinical issues and to be able to break information down [8], a high level of understanding of the disease is needed in the Scholars180 assessment to make judgments about core pieces of information. In addition, the short presentation style cuts down the length of time required to complete the assessment, which is often an issue with oral assessments [6]. Moreover, the public presentation style, rather than an online or self-record version, has been chosen as it has been shown that students tend to take presenting in front of others more seriously and thus are more likely to study the topic in depth in comparison to if they weren't presenting publicly [9].

After the presentation, the students were required to answer three questions in an oral format (worth 3% of the total mark for the unit) in a separate room with an academic about the same eye condition. The questions were based on existing clinical behavior patterns in eye clinics (clinical workup) and the required approaches to ensure patients receive evidence-based care. The questions were as follows:

1. How would you test for this disease and what clinical workup would you use?

**Table 1. List of eye conditions.**

| Number | Ocular Disease |
|---|---|
| 1. | Branch Retinal Vein Occlusion |
| 2. | Pseudoexfoliative Glaucoma (PXF) |
| 3. | Herpes Simplex Keratitis |
| 4. | Keratoconus |
| 5. | Microbial Keratitis |
| 6. | Trachoma |
| 7. | Thyroid Eye Disease |
| 8. | Orbital Cellulitis |
| 9. | Ocular Albinism |
| 10. | Ocular Allergy |
| 11. | Uveitis |
| 12. | Hypertensive Retinopathy |
| 13. | Pigment Dispersion Syndrome |
| 14. | Iris Coloboma |
| 15. | Wet ARMD |
| 16. | Dry ARMD |
| 17. | Diabetic Retinopathy |
| 18. | Cataract |
| 19. | Orbital Blow Out Fracture |
| 20. | Strabismic Amblyopia |
| 21. | Colour Vision Deficiency |
| 22. | Optic Neuritis |
| 23. | Pituitary Tumour |
| 24. | Primary Open Angle Glaucoma |
| 25. | Horner's Syndrome |
| 26. | Scleritis |
| 27. | Basal Cell Carcinoma of the Eyelid |
| 28. | Nasolacrimal Obstruction |
| 29. | Meibomian Gland Dysfunction |
| 30. | Choroidal Naevus |
| 31. | Retinal Detachment |
| 32. | Recurrent Corneal Erosion |
| 33. | Acute Angle Closure |
| 34. | Pterygium |
| 35. | Central Retinal Artery Occlusion |
| 36. | Squamous Cell Carcinoma |
| 37. | Central Retinal Artery Occlusion |
| 38. | Squamous Cell Carcinoma |
| 39. | Marcus Gunn Phenomenon |
| 40. | Corneal Abrasion |
| 41. | Trichiasis |
| 42. | Conjunctivitis |
| 43. | Chemical Burn |
| 44. | Retinitis Pigmentosa |
| 45. | Blepharitis |

2. Pretend you are in the clinic, and you suspect a patient has this disease, how would you manage the patient?

3. What did you learn from this assignment?

The Q&A component was included to improve the authenticity of the assessment. Authenticity refers to how the assessment mimics encounters in the workplace or real-life situations [10]. This allowed the teacher to probe questions from the student to test whether their understanding was superficial or in-depth [2, 11, 12]. The questions asked required clinical reasoning and students need to critically think about how to approach the question to answer it correctly. This is beneficial as in clinical practice, optometrists are faced with many different cases in which the application of knowledge is needed and not direct regurgitation of information.

Before any tests were conducted, the students were assigned a uniquely identifiable number that they would have to use for both the pre- and post-questionnaires. The spreadsheet with the unique identification numbers was stored on a password-protected computer and the extracted information from the spreadsheet did not include any details that could be used to identify the students (to ensure anonymity).

To test the hypothesis, this study involved an anonymous pre-questionnaire data collection to measure the students' knowledge of eye diseases and their understanding of the scope of practice, followed by the development and sharing of the presentations and a subsequent post-questionnaire to evaluate their change in knowledge and behavior. The pre- and post-questionnaires were useful as the baseline assessment of knowledge (pre-questionnaire) primed students for learning as it made them aware of what they did not know. The questions for the pre-questionnaire of existing knowledge can be found in S2 Appendix and the students answered the questions on a 5-point Likert scale. Depending on the question, the 5-point Likert scale score included: (1) Very low/strongly disagree/very unlikely; (2) Low/disagree/unlikely; (3) Medium/neutral; (4) High/agree/likely: (5) Very high/strongly agree/very likely. The questions were divided into three domains including knowledge of eye diseases, attitude and intention towards learning, and application of knowledge to placement (practice). For each domain, specific questions were aimed to test an aspect of that domain.

After the delivery of the presentations, the students were then asked to complete a post-questionnaire found in S3 Appendix to measure a difference in knowledge and or behavior. The post-questionnaire also supported student reflection, which consolidated learning. As part of the evaluation test, besides the questions on knowledge, attitude, intentions, and practice, the students were asked about the performance of their peers too, which is shown in S4 Appendix.

Data from participants who completed both pre- and post-questionnaires were included for data analysis. All statistics were performed in R v4.1.0. Descriptive statistics were used to report demographic data. Non-parametric Wilcoxon Signed-Ranks Tests (with continuity correction) were used to compare the responses to each of the 12 Likert-type scale questions on the post-questionnaire with the respective responses on the pre-questionnaire. To account for multiple comparisons, a Bonferroni correction was applied to the initial alpha of 0.05. Given that 12 statistical tests were applied, the adjusted alpha (via Bonferroni correction) was calculated to be 0.004; this means statistical significance was deemed to be achieved if the resultant $P$-values were less than 0.004.

## Results

Overall, as shown in Table 2, 42 students completed the pre-oral presentation questionnaire, and 41 completed the post-questionnaire. However, only 31 completed both pre- and post-questionnaires as incomplete surveys were removed from the data.

**Table 2. Demographic characteristics of test participants (n = 31).**

| Characteristics | | n (%) |
|---|---|---|
| Sex | | |
| | Male | 4 (13) |
| | Female | 27 (87) |
| Age in years | | |
| | 18–24 | 26 (84) |
| | 25–34 | 5 (16) |
| | 35–44 | 0 (0) |
| | 45 and over | 0 (0) |
| Highest level of formal education | | |
| | Bachelor | 28 (90) |
| | Bachelor with honors | 3 (10) |
| | Master | 0 (0) |
| | Ph.D. or higher | 0 (0) |

The results as shown in Table 3 indicated how the Scholars180 assessment impacted student knowledge of ocular diseases and how confident they were in the identification and management of these diseases. Generally, Table 3 showed that the post-questionnaire median scores for the statements increased from the pre-questionnaire scores. For the 'knowledge of eye diseases' domain, all values increased. The median values for overall knowledge, knowledge of

**Table 3. Self-reported knowledge, attitudes, and knowledge application pre- and post-oral presentation (n = 31)[a].**

| Domain | | Pre Median (IQR) | Post Median (IQR) | Median change | P-value[b] |
|---|---|---|---|---|---|
| Knowledge of eye diseases | | | | | |
| | Overall knowledge | 2.0 (1.0) | 3.0 (1.0) | +1.0 | < .001[b] |
| | Common treatment methods | 2.0 (1.0) | 3.0 (1.0) | +1.0 | < .001[b] |
| | Diagnostic knowledge | 2.0 (1.0) | 3.0 (1.0) | +1.0 | < .001[b] |
| Attitude and intention | | | | | |
| | Assessment methods need improvement | 3.0 (1.0) | 3.0 (1.0) | 0.0 | >.99 |
| | Confident about signs | 2.0 (1.0) | 4.0 (1.0) | +2.0 | < .001[b] |
| | Confident about symptoms | 2.0 (1.0) | 4.0 (1.0) | +2.0 | < .001[b] |
| | Confident about the treatment options | 2.0 (1.0) | 3.0 (1.0) | +1.0 | < .001[b] |
| | Confident about when to manage or refer a patient | 2.0 (1.0) | 4.0 (1.0) | +2.0 | < .001[b] |
| Application of knowledge to placement (practice) | | | | | |
| | Appropriate management of chronic eye conditions | 1.0 (1.0) | 3.0 (0.0) | +2.0 | < .001[b] |
| | Appropriate management of acute eye conditions | 1.0 (1.0) | 3.0 (0.0) | +2.0 | < .001[b] |
| | Appropriate management of severe eye conditions | 1.0 (1.0) | 3.0 (0.5) | +2.0 | < .001[b] |
| | Appropriate management of minor eye conditions | 2.0 (1.0) | 3.0 (1.0) | +1.0 | < .001[b] |

[a]Scores for each item ranged from 1 to 5 with higher scores indicating a higher level of self-reported knowledge, positive attitude, and/or knowledge application.

[b]Indicates statistically significant change from pre- to post-oral presentation, given Bonferroni corrected P-value cut-off of 0.004

common treatment methods, and diagnostic knowledge all increased from 2.0 to 3.0. Concerning the attitudes and intention domain, all but one of the median scores increased. Student confidence regarding the signs and symptoms of ocular diseases and when to manage or refer a patient all improved as the median scores increased from 2.0 to 4.0. Student confidence about the treatment options available also improved from a score of 2.0 to 3.0. However, for the statement regarding "assessment methods need improvement", the score remained at 3.0 for the pre- and post-questionnaire. With regards to the 'application of knowledge to placement' domain, all median scores increased. There was a significant improvement in the appropriate management of chronic eye conditions, acute eye conditions, and severe eye conditions as the median scores increased from 1.0 to 3.0. Appropriate management of minor eye conditions also increased from 2.0 to 3.0. Furthermore, not only did median scores increase in the post-questionnaire, but they also increased by a *P*-value below 0.001. Additionally, the interquartile range (IQR) for all the statements in Table 2 was 1.0 or lower. This indicated that the spread of data was minimal, which suggested that the students had similar experiences with the assessment.

Moreover, the largest changes in scores were mainly regarding the student's confidence in the identification and management of ocular diseases. For example, 3 out of 5 statements under the attitude and intention domain had a median change of an increase of 2.0 points. The increase of 2.0 points on the median score also applied to 3 out of 4 of the statements on the application of knowledge to the placement domain. In comparison, for all three statements in the knowledge of eye diseases domain, the median score increased by one, which was a lesser amount than the other domains. Overall, the results indicated that the median student attitude towards ocular diseases improved in a positive direction as their answers to the statements changed from 'disagree' to 'neutral' or 'agree'.

Table 4 illustrated student evaluation of their peers' presentations. For all the statements in Table 4, the median score was 4.0. out of a maximum score of 5.0. The 4.0 scores for statements such as 'the shared knowledge was relevant to my growth as a student', 'I would recommend this assessment to examine future students', 'overall, I was satisfied with the processes of the assignment' etc. showed that not only did students benefit from presenting themselves but also when their peers did. Additionally, the IQR for all the statements was 1.0 or lower, indicating that most students had a high level of agreement regarding the statements on peer evaluation. It is important to note that the three questions asked after the oral presentation were relevant in helping the students reflect on their learning. This allowed them to answer the questions more accurately under the domains of 'knowledge of eye diseases' and 'application of knowledge to placement (practice)'.

**Table 4. Peer evaluation of assignment organization (n = 31)[c].**

| Statements | Median (IQR) |
|---|---|
| Most of the presentations were visually appealing | 4.0 (0.5) |
| The shared knowledge was relevant to my growth as a student | 4.0 (1.0) |
| I understood most of the terminologies used in the oral presentations of my classmates | 4.0 (1.0) |
| The content of the presentations met my expectations | 4.0 (1.0) |
| I would recommend this assessment to examine future students | 4.0 (1.0) |
| Overall, I was satisfied with the processes of the assignment | 4.0 (1.0) |

[c]Scores for each item ranged from 1 to 5 with higher scores indicating a higher level of agreement with the statements.

## Discussion

This study's goal was to study the impact of the oral presentation assessment, Scholars180, on optometry students' knowledge of ocular diseases and their confidence regarding how to identify and manage these diseases. Oral assessments are defined as an assessment of student learning that incorporates spoken words either entirely or partially [11]. One of the important competencies that health professionals need to learn is the ability to effectively articulate their medical knowledge to patients of varying backgrounds [13]. To do this, the health professional must have sufficient knowledge of the disease and be able to identify it, but they also must be able to communicate all relevant aspects of the disease to the patient within short consultation times, so patients can play an active role in their health decisions. Oral assessments are multi-functional assessments that can both evaluate and assist in students achieving several learning outcomes. Hence, after completion of Scholars180, it was hypothesized that student knowledge of and confidence in the identification and management of ocular diseases will improve. In addition, the peer learning aspect of Scholars180 may have also contributed to an improved learning experience.

From the results as indicated in Table 2, our hypothesis was supported as the median scores increased by a statistically significant value with a low IQR. This illustrated that students felt that their knowledge of and confidence to identify and manage ocular diseases had improved. The data collected is encouraging as it revealed that the design of Scholars180 is in the correct direction. To complete the assessment, students needed to compress all the relevant information about an ocular disease in a very limited time, which required them to gain an in-depth understanding of the topic to differentiate between essential and non-essential pieces of information and how to string it together so it could be easily understood. Furthermore, due to the time constraint, the 3MT format encouraged students to be succinct yet engaging to make an impression on the audience [14]. Therefore, due to the high demands for successful completion of Scholars180, it was likely that students were pushed to gain in-depth knowledge of ocular diseases, which transferred to an increase in confidence in identification and management as well.

Moreover, the results shown in Table 3 highlighted how students felt about Scholars180 after completing the assessment. From Table 3, it is seen that the median scores were all 4.0, indicating the students mostly agreed with the statements and thus felt positively towards the assessment. This is most likely due to the benefits discussed above, but it may also be attributed to the peer learning aspect of Scholars180. Peer learning can be defined simply as the interaction between students to actively learn educational material [15]. In Scholars180, peer learning was present as students were actively teaching other students about their assigned ocular disease, and students also learned from their peer's presentations. The benefits of the teaching aspect of oral presentations are consolidated by a study that found that teaching increases reflection, breaks down resistance to change, and can help one to recognize their ignorance, leading them to be more open to learning [16]. When expected to teach content, students are more likely to be engaged with the information. This is likely because the process of preparing for oral presentations required students to first practice with self-explanation which could expose gaps in their knowledge, leading them to seek further studying to gain a deeper level of understanding [17]. Moreover, by listening to other presentations, students could reinforce their knowledge regarding each ocular disease [18].

Moving forward, assessments can be summative or formative [19]. Summative assessments are usually carried out at the end of a learning process to test if a student has sufficiently learned the material and it tends to be a high-stakes assessment [3]. In contrast, formative assessments are usually conducted during the student's learning process to provide them with

feedback and the outcome does not have a high impact on the student's final score [3]. For effective evaluation, formative assessments should be designed to help strengthen summative assessments [3]. Scholars180 is a summative assessment as it is conducted at the end of the semester and accounts for 7% of the student's final grade in the unit. The data collected suggests that it is an effective summative assessment. However, in the future, it could be helpful to design formative assessments to accompany it throughout the semester as obtaining relevant and early feedback is an important aspect of facilitating improvement [20, 21] and a study has shown that this is particularly true for low-performing students [22]. For example, a study found that one-minute presentations conducted every few weeks to provide students with immediate feedback helped them develop better clarity in speaking and confidence in presenting [23].

There are also some limitations to the design of the study. The participants conducted the pre-questionnaire at the beginning of the course whilst the post-questionnaire was conducted 12 weeks later. As there is a time gap between the questionnaires, it is unsure if the increase in student knowledge and confidence is due to the Scholars180 assessment itself, or through other teaching activities such as lectures, tutorials, and tests. Moreover, the sample size for this study was relatively small and there was no comparator group given a standard oral presentation assessment task. Thus, it is not known whether the improvements that students felt from Scholars180 would fare better than a standard oral assessment. In the future, a large sample should be used to permit more generalisability of the outcomes. The differences in outcomes for first-year students may be due to their different educational backgrounds and exposure to presentation assessments. Hence, this study could be conducted on second or third-year optometry students once they have all been exposed to this assessment type, which enables a more standardised comparison of results.

Lastly, the students were surveyed using a 5-point Likert scale regarding their confidence and self-reported knowledge of subjects. However, the rating is quite subjective and there is no way of knowing if their reported increase in confidence on the subject translated to an actual increase in knowledge. Another detail to note is that during the development of the project, a gap in knowledge around the establishment of clear and comprehensive rubrics for oral presentations was identified. To fill this knowledge gap, various journals on rubric development and consultations with established academics were used to improve the development of new assessment tools and associated rubrics. However, more research in the future concerning rubric design could help refine Scholars180.

Given that Scholars180 is an effective assessment, students may benefit directly from participation by gaining better knowledge in the areas of optometry, ophthalmology, and vision sciences. There is also a chance that some of the students may not directly benefit from this assignment. However, the results of their tests will help guide future research into this area as well as provide direction for improvements in how we teach and assess optometry students. Doing so could help improve the health of the wider community by improving the competence of the optometry workforce.

## Conclusion

In this study, the impact of Scholars180, an oral presentation assessment tool, has been analyzed and discussed. The results have supported the hypothesis that Scholars180 would help students improve their knowledge of ocular diseases and their confidence in the identification and management of the disease. It is also likely that students would benefit from the peer learning aspect of the assessment. Scholars180's oral presentation assessment design provided a method to shorten the time required for typical oral presentation assessments whilst

simultaneously maintaining the effectiveness of the assessment. Overall, the design of Scholars180 appeared to be in the right direction, but further research should be conducted to strengthen the assessment tool. Through this study, an additional oral presentation assessment and its benefits to students will be added to the existing literature. Additionally, it may also encourage other academics to consider implementing this assessment or aspects of it in their curriculum to enhance both the teaching and learning experience.

## Supporting information

**S1 Appendix. Rubric for the assignment.**
(DOCX)

**S2 Appendix. Questions for pre-questionnaire.**
(DOCX)

**S3 Appendix. Questions for post-questionnaire.**
(DOCX)

**S4 Appendix. Questions for peer evaluation.**
(DOCX)

**S1 File. Scholars180 pre-questionnaire raw data.**
(XLSX)

**S2 File. Scholars180 post-questionnaire raw data.**
(XLSX)

## Author Contributions

**Conceptualization:** Khyber Alam, Kenneth Lee.

**Data curation:** Khyber Alam, Kate Vance.

**Formal analysis:** Khyber Alam.

**Investigation:** Khyber Alam.

**Methodology:** Khyber Alam.

**Writing – original draft:** Khyber Alam, Xinyi Zheng.

**Writing – review & editing:** Khyber Alam, Xinyi Zheng, Kenneth Lee.

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
