## [Decision Letter · Decision Letter 0]

24 May 2023

PONE-D-23-06339Scholars180: An effective oral presentation assessment for optometry students that overcomes time limitationsPLOS ONE

Dear Dr. Zheng,

Thank you for submitting your manuscript to PLOS ONE. After careful consideration, we feel that it has merit but does not fully meet PLOS ONE’s publication criteria as it currently stands. Therefore, we invite you to submit a revised version of the manuscript that addresses the points raised during the review process.

We look forward to receiving your revised manuscript.

Kind regards,

Kofi Asiedu

Academic Editor

PLOS ONE

Journal Requirements:

Reviewers' comments:

Reviewer's Responses to Questions

**Comments to the Author**

1. Is the manuscript technically sound, and do the data support the conclusions?

Reviewer #1: Yes

Reviewer #2: Yes

2. Has the statistical analysis been performed appropriately and rigorously? 

Reviewer #1: Yes

Reviewer #2: Yes

3. Have the authors made all data underlying the findings in their manuscript fully available?

Reviewer #1: Yes

Reviewer #2: Yes

4. Is the manuscript presented in an intelligible fashion and written in standard English?

Reviewer #1: Yes

Reviewer #2: Yes

5. Review Comments to the Author

Reviewer #1: The research work is very relevant in adding to the available data on summative assessment. The Scholars180 is a very good tool that can rigorously be used as a research tool in checking the effectiveness of oral presentation in the assessment of students. The authors provided pieces of information on the baseline knowledge of the study participants, the pre-presentation questionnaires, 180 seconds presentation, question and answers and the post presentation questionnaires which are very vital in proving the effectiveness of the presentation tool used in this study which is Scholars180.

The research work employed the right methodology. The authors did well in using questionnaires before the presentation and after the presentation. The non-parametric test employed in this research work is well suited for such purpose. The conclusion made by the authors helps in contributing to the available data on the usefulness and effectiveness of oral presentations in the assessment of students.

However, in the future, the work has to be done on a larger sample to permit for more generalizability of the outcomes. Also, in the future the research can also be conducted in those in either the second or third year since they might have had exposure to presentations which will put all participants at the same level. In such cases then the effectiveness of the Scholars180

Reviewer #2: General comment:

The manuscript concept is essential to the academic evaluation of students in all faculties, not only in optometry. Various techniques, such as direct observation abilities, standardized interpretation training, and having an expert double-check, have been suggested to evaluate learners' image interpretation. Oral reporting, essays, feedback, or straightforward multiple-choice questions are frequently used to assess a learner's performance. Still, unless these tools are well-designed, it may not be easy to discern a learner's clinical reasoning process. Scholars180 is a novel assessment tool for evaluating students and will have great potential in its impact on optometry training.

Comments

1. The title of the manuscript could be modified since the manuscript did not evaluate time limitations. Suggested title: “Scholars180: An effective oral presentation assessment for Optometry Students.”

2. There was mention of three questions used in assessing the presenters in a separate room. The manuscript needs to cover that in the results and discussion section.

3. The follow-up time for the measurement of the post-questionnaire should have been reported in the methodology.

4. If there was one, the manuscript did not state the sampling process and exclusion.

5. From the manuscript, lines 111-117 and 149-154 seem to repeat the same concept. It is my opinion to omit one.

6. In lines 196-197, you stated the Likert scale used in Appendix 2, but Q5 and Q7 used different wording of the scale. I think using the phrasing of a “5-point Likert scale score” only without itemizing them would be appropriate.

7. The question domain at line 198, “attitude and intention,” is unclear if the attitude concerns eye diseases or the learning process. As shown in the appendix, line 433, it is suggested to use “Attitude and intention towards learning.”

8. Were the pre and post-questionnaire results based on all the topics in the different presentations or on a single presentation based on a single eye condition? Can that be clarified in the manuscript?

9. The Likert scale for the question domain knowledge about eye disease could have been: (1 = very unconfident, 2 = fairly unconfident, 3 = neutral, 4 = fairly confident, and 5 = very confident) instead of high to low.

10. The Likert scale for the question domain practice “likely,” reads unethical to be used for questions on patient management. I believe a different wording could have been used, like “confident.”

11. The limitations presented in the manuscript are the main defining pointers to measure if the Scholar180 assessment was the primary influencer for the increase in knowledge of the peers. These limitations could have been taken care of to objectively measure the benefit of using this assessment method.

6. PLOS authors have the option to publish the peer review history of their article (what does this mean?). If published, this will include your full peer review and any attached files.

Reviewer #1: No

Reviewer #2: No

---

## [Author Response · Author response to Decision Letter 0]

27 Jun 2023

Firstly, we would like to thank the reviewers for their time and effort in reviewing our manuscript. The constructive criticism provided in the comments has enabled us to improve the quality of our work. We have revised our manuscript according to most of the comments provided. The changes have been highlighted in red in the ‘revised manuscript with tracked changes’ document as requested and our responses are given below. Our response to the reviewers is written below, with the direct comments by the reviewers coloured in blue and italicised. The referred lines correspond to the revised manuscript, not the first submission unless stated otherwise. 

The journal requirements have been addressed. The layout of the manuscript has been changed to fit the templates provided by PLOS One. The section ‘Supporting Information’ has been added to include the appendixes and the minimal dataset underlying the results has been included as Supporting Information files (S1 File and S2 File). The ethics statement has also been moved to the methodology section (lines 134-142). 

Responses to Reviewer 1:

Thank you to reviewer 1 for the positive feedback provided. 

Comment 1: However, in the future, the work has to be done on a larger sample to permit for more generalizability of the outcomes. Also, in the future the research can also be conducted in those in either the second or third year since they might have had exposure to presentations which will put all participants at the same level. In such cases then the effectiveness of the Scholars180

Response 1: The authors acknowledge that this study can be improved by using a larger sample size and with second and third-year optometry students. Hence, we have added these points raised in our limitations paragraph in the discussion section as seen in lines 365-366 and 368-374.

Responses to Reviewer 2: 

We appreciate your acknowledgement of the benefits of using Scholars180 as an assessment tool. 

Comment 1: The title of the manuscript could be modified since the manuscript did not evaluate time limitations. Suggested title: “Scholars180: An effective oral presentation assessment for Optometry Students.”

Response 1: We agree with the comment and the suggested title has been adopted to better reflect the paper’s contents. 

Comment 2: There was mention of three questions used in assessing the presenters in a separate room. The manuscript needs to cover that in the results and discussion section.

Response 2: The three questions and the relevant context has been appropriately explained in the methodology and we added some information to the results section to emphasise its relevance as seen in line 282-284. The questions were part of the overall assessments, and we believe additional information purely focused on the questions will influence the overall flow of the discussion section. 

Comment 3: The follow-up time for the measurement of the post-questionnaire should have been reported in the methodology.

Response 3: The follow-up time has now been reported in the second paragraph, lines 144-146, in the methodology section as 12 weeks. 

Comment 4: If there was one, the manuscript did not state the sampling process and exclusion.

Response 4: There was no sampling or exclusion process. All first-year optometry students at the University of Western Australia were given the opportunity to participate in the study. The students were only excluded if they did not consent to participate. We apologise for not making this clear in the study and have included a sentence to make this clear in the first paragraph of the methodology section (lines 139-140). Additionally, the results of some students were excluded due to incomplete questionnaires, which was stated in the first paragraph, lines 241-243, of the results section. 

Comment 5: From the manuscript, lines 111-117 and 149-154 seem to repeat the same concept. It is my opinion to omit one.

Response 5: We acknowledge the concepts are the same. A sentence in lines 157-158 has been removed to prevent repetition. The deletion of this also removed the reference by Hartigan L (originally the 8th reference) as it is no longer essential to the paper. However, we believe the whole of lines 149-154 of the original paper cannot be removed as lines 149-152 explain the 3-minute thesis format required in the methodology. Lines 165-168 of the revised manuscript are kept as it links to the other ideas in the paragraph smoothly. Additionally, although the two concepts are the same, they are in separate sections- the introduction and methodology. We chose to keep lines 111-117 of the original manuscript in the introduction as we believe it is necessary to introduce the uniqueness of the three-minute thesis format as it is not widely known in other countries besides Australia. 

Comment 6: In lines 196-197, you stated the Likert scale used in Appendix 2, but Q5 and Q7 used different wording of the scale. I think using the phrasing of a “5-point Likert scale score” only without itemizing them would be appropriate.

Response 6: Thank you for pointing out the inconsistencies. We have included the other descriptors for questions 5 and 7 in the methodology as seen in lines 209-212. In line 375, we have also amended the phrasing to not be itemized. 

Comment 7: The question domain at line 198, “attitude and intention,” is unclear if the attitude concerns eye diseases or the learning process. As shown in the appendix, line 433, it is suggested to use “Attitude and intention towards learning.”

Response 7: This has been amended to “attitude and intention towards learning” in now line 213.

Comment 8: Were the pre and post-questionnaire results based on all the topics in the different presentations or on a single presentation based on a single eye condition? Can that be clarified in the manuscript?

Response 8: This has been clarified in the methodology section in lines 143-144. The pre and post-questionnaires were based on all the topics in the different presentations.

Comment 9: The Likert scale for the question domain knowledge about eye disease could have been: (1 = very unconfident, 2 = fairly unconfident, 3 = neutral, 4 = fairly confident, and 5 = very confident) instead of high to low.

Response 9: We agree that using ‘confident’ rather than ‘high to low’ would have been a better option as ‘high to low’ can be somewhat subjective. This is an aspect that we will need to be more careful of when considering the terminology we use in our scales. However, as the result included measurements of median changes, we believe that it still sufficiently displays the benefits of Scholars180 for the student’s development. 

Comment 10: The Likert scale for the question domain practice “likely,” reads unethical to be used for questions on patient management. I believe a different wording could have been used, like “confident.”

Response 10: The word ‘confident’ could have been a better wording option than ‘likely’ and we will keep that in mind for future studies that involve the Likert scale and questions regarding patient management. However, we also believe the word ‘likely’ cannot be considered ‘unethical’ in this scenario as the study is conducted on first-year optometry students. Hence, there is no expectation for the students to be competent in patient management as the goal is to achieve the skills through participating in the assessment. 

Comment 11: The limitations presented in the manuscript are the main defining pointers to measure if the Scholar180 assessment was the primary influencer for the increase in knowledge of the peers. These limitations could have been taken care of to objectively measure the benefit of using this assessment method.

Response 11: We acknowledge that some of the limitations could have been better-taken care of. However, we do not believe that the limitations are significant enough to detract from showing the positive benefits of implementing a novel oral presentation like Scholars180.

---

## [Decision Letter · Decision Letter 1]

11 Jul 2023

Scholars180: An effective oral presentation assessment for optometry students

PONE-D-23-06339R1

Dear Dr. Zheng,

We’re pleased to inform you that your manuscript has been judged scientifically suitable for publication and will be formally accepted for publication once it meets all outstanding technical requirements.

Kind regards,

Kofi Asiedu

Academic Editor

PLOS ONE

Additional Editor Comments (optional):

Reviewers' comments:

Reviewer's Responses to Questions

**Comments to the Author**

1. If the authors have adequately addressed your comments raised in a previous round of review and you feel that this manuscript is now acceptable for publication, you may indicate that here to bypass the “Comments to the Author” section, enter your conflict of interest statement in the “Confidential to Editor” section, and submit your "Accept" recommendation.

Reviewer #1: All comments have been addressed

Reviewer #2: All comments have been addressed

2. Is the manuscript technically sound, and do the data support the conclusions?

Reviewer #1: Yes

Reviewer #2: Yes

3. Has the statistical analysis been performed appropriately and rigorously? 

Reviewer #1: Yes

Reviewer #2: Yes

4. Have the authors made all data underlying the findings in their manuscript fully available?

Reviewer #1: Yes

Reviewer #2: Yes

5. Is the manuscript presented in an intelligible fashion and written in standard English?

Reviewer #1: Yes

Reviewer #2: Yes

6. Review Comments to the Author

Reviewer #1: (No Response)

Reviewer #2: The authors have effectively addressed and revised the manuscript to address the concerns raised adequately and I agree that it can be considered and accepted for publication.

7. PLOS authors have the option to publish the peer review history of their article (what does this mean?). If published, this will include your full peer review and any attached files.

Reviewer #1: No

Reviewer #2: No

---

## [Editor Report · Acceptance letter]

14 Jul 2023

PONE-D-23-06339R1 

Scholars180: An effective oral presentation assessment for optometry students 

Dear Dr. Zheng:

I'm pleased to inform you that your manuscript has been deemed suitable for publication in PLOS ONE. Congratulations! Your manuscript is now with our production department. 

Kind regards, 

on behalf of

Dr. Kofi Asiedu 

Academic Editor

PLOS ONE